# Comparison of Two Endoscopic Therapeutic Interventions as Primary Treatment for Anastomotic Leakages after Total Gastrectomy

**DOI:** 10.3390/cancers14122982

**Published:** 2022-06-16

**Authors:** Moritz Senne, Christoph R. Werner, Ulrike Schempf, Karolin Thiel, Alfred Königsrainer, Dörte Wichmann

**Affiliations:** 1Department of General, Visceral and Transplantation Surgery, University Hospital Tübingen, Hoppe-Seyler-Str. 3, 72076 Tübingen, Germany; moritz.senne@med.uni-tuebingen.de (M.S.); karolin.thiel@med.uni-tuebingen.de (K.T.); alfred.koenigsrainer@med.uni-tuebingen.de (A.K.); doerte.wichmann@med.uni-tuebingen.de (D.W.); 2Department of Gastroenterology, Gastrointestinal Oncology, Hepatology, Infectiology, and Geriatrics, University Hospital Tübingen, Otfried-Müller-Str. 10, 72076 Tübingen, Germany; ulrike.schempf@med.uni-tuebingen.de

**Keywords:** esophagojejunal anastomotic leak, oncological gastrectomy, complication management, endoscopy

## Abstract

**Simple Summary:**

An esophagojejunal anastomotic leak after oncological gastrectomy is a life-threatening complication. Endoscopic treatment offers the possibility of minimally invasive diagnosis and immediate effective therapy in one session. A retrospective, single-center analysis of two different endoscopic strategies as first-line treatment options was performed.

**Abstract:**

Introduction: An esophagojejunal anastomotic leak following an oncological gastrectomy is a life-threatening complication, and its management is challenging. A stent application and endoscopic negative pressure therapy are possible therapeutic options. A clinical comparison of these strategies has been missing until now. Methods: A retrospective analysis of 14 consecutive patients endoscopically treated for an anastomotic leak after a gastrectomy between June 2014 and December 2019 was performed. Results: The mean time of the diagnosis of the leakage was 7.14 days after surgery. Five patients were selected for a covered stent, and nine patients received endoscopic negative pressure therapy. In the stent group, the mean number of endoscopies was 2.4, the mean duration of therapy was 26 days, and the mean time of hospitalization was 30 days. In patients treated with endoscopic negative pressure therapy, the mean number of endoscopies was 6.0, the mean days of therapy duration was 14.78, and the mean days of hospitalization was 38.11. Treatment was successful in all patients in the stent-based therapy group and in eight of nine patients in the negative pressure therapy group. Discussion: Good clinical results in preserving the anastomosis and providing sepsis control was achieved in all patients. Stent therapy resulted in anastomosis healing with a lower number of endoscopies, a shorter time of hospitalization, and rapid oral nutrition.

## 1. Introduction

Gastric cancer is one of the leading causes of cancer-related deaths in the world, and multimodal treatment combining perioperative chemotherapy with radical resection and D2 lymphadenectomy is the established curative treatment [1]. For a subtotal and a complete gastrectomy, digestive reconstruction is performed with a Roux-en-Y or jejunal interposition; both include anastomosis between the esophagus/stomach and the jejunum. Esophagojejunal anastomotic leakage (EJAL) is one of the most serious complications, with an incidence between 0.5% and 11.0%, and is associated with a high mortality rate as well as longer ICU and hospital stays [2,3,4]. For advanced but resectable gastric carcinoma, multimodal therapy has been established with preoperatively neoadjuvant chemotherapy, which should be completed postoperatively for an improved rate of disease-free survival [5]. EJAL is likely to delay (or render impossible) adjuvant chemotherapy, resulting in a poorer oncological treatment [6,7]. Therefore, therapy for EJAL must consider not only the treatment of acute complications, such as sepsis, the inability to receive enteral nutrition, and respiratory failure, but also the impact on multimodal oncological therapy. Therapeutic options include conservative treatment with AB, interventional endoscopy, and re-operation. Possible endoscopic options include stenting, clipping, and endoscopic intraluminal negative pressure therapy (ENPT). The choice of treatment is made according to the patient’s clinical condition, timing of diagnosis, anastomotic level, size of the leakage, and perfusion of the jejunum. However, there are no valid guidelines available for the treatment of EJAL.

The placement of self-expandable metal stents is carried out to cover the insufficiency and to prevent the extravasation of secretions into the para-anastomotic room, enabling oral feeding. Para-anastomotic collections have to be drained either by percutaneous drain or re-operation. The stent may be left in place for six to eight weeks and can be removed endoscopically at the end of therapy [8].

Clipping is an effective and elegant treatment for early, small insufficiencies with good perfusion of the margins and a missing cavity outside. As with stent therapy, oral feeding is possible immediately after intervention without the need for re-endoscopy to remove the clipping material.

A third endoscopic treatment option is ENPT. This technique has become a promising tool for managing complications after surgery in both the upper and lower gastrointestinal tract [9,10,11,12,13]. ENPT improves local perfusion, the resolution of interstitial wound edema, the removal of fluids, and the debridement of the wound ground [14,15,16]. Vital granulation tissue is formed after wound cleaning and when endoluminal or intracavitary ENPT are possible. Closed re-endoscopies and a lack of oral nutrition are negative aspects of this therapy. Enteral feeding requires a deep nasojejunal tube, which limits patient comfort and, along with the vacuum pump and repeat endoscopies, may be a reason for a prolonged hospital stay. Only a very carefully selected patient population is suitable for outpatient ENPT therapy [17].

Since 2018, ENPT has been the primary endoscopic therapeutic option for EJAL at our institution, after years of stenting using different self-expandable metal stents (SEMSs). The reason for changing our approach to primary ENTP for EJAL was better wound healing with simultaneous sepsis control and a lower re-operation rate. Furthermore, the treated patients retained the possibility of receiving adjuvant chemotherapy according to clinical recommendations. The aim of this study was to clinically compare two treatment strategies in a 5-year period.

## 2. Materials and Methods

### 2.1. Study Design

The local ethics committee of Tübingen University Hospital, Germany, approved this study (AZ: 752/2019BO2, date of approval 31 October 2019). The study is registered with ClinicalTrials.gov under the reference number NCT04362605. This study adheres to the PROCESS guidelines from 2016.

All EJAL patients primarily treated with stent placement or endoscopic negative pressure therapy between January 2014 and December 2019 were considered for this retrospective study. Inclusion criteria were adult patients after elective gastrectomy with a curative approach with an EJAL and primary endoscopic intervention between 2014 and December 2019. Patients in a palliative setting and with primary surgical re-do anastomosis after diagnosis of EJAL were excluded. As a primary outcome parameter, we investigated the success rate of endoscopic interventions on the healing of the anastomosis. Secondary outcomes were needed for invasive ventilation, time in ICU, therapy duration, possibility of oral and enteral feeding, number of interventions (surgery and endoscopy), and length of hospital stay.

All patients who underwent primarily endoscopic therapy for EJAL before 2018 were treated with stents. After 2018, they were treated with ENPT. The decision of endoscopic primary therapy was made by the operating surgeon and the interventional endoscopist. The reason for changing our approach from stent-based therapy to primary ENTP for EJAL was based on the results of ENPT for various gastrointestinal defects.

Diagnostic and therapeutic decision making is shown in Figure 1 as a flowchart. In accordance with ethical approval, informed consent was obtained from all participants. Patients’ records, as well as the database, were analyzed for EJAL therapy-specific items.

### 2.2. Procedural Information

For all patients, the surgical procedure of the esophagojejunostomy for oncological gastrectomy or combined transhiatal gastrectomy was performed end to side with a circular stapler, without a jejunal pouch. The blind end of the jejunum was closed with a linear stapler.

In patients with suspected EJAL, first, an index endoscopy was performed after informed consent was obtained, under general anesthesia, with endotracheal intubation as well as a CT scan to detect extraluminal fluid or abscess. Standard gastroscopes with an outer diameter of 9.8 mm were used with carbon dioxide insufflation. The definition of an anastomotic leak was based on the endoscopic finding in the esophagojejunostomy according to the CAES classification [18]. Extraluminal collections were drained percutaneously or via a limited surgical procedure. Therapeutic success was defined as complete resolution of the perforation.

### 2.3. Endoscopic Stent Placement

A nitinol fully covered Wallflex^®^ stent (Boston Scientific, Natick, MA, USA) with the dimensions of 105 mm in length and 23 mm in diameter was used in all analyzed patients. Implantation was realized under endoscopic visualization without X-ray. An endoscopic control is usually performed after 4–6 weeks for stent removal or re-implantation of a new stent. If stent dislocation was suspected because of clinical symptoms such as nausea and vomiting or dysphagia, re-endoscopy was performed earlier. End of treatment was defined after removal of the stent.

### 2.4. Endoscopic Negative Pressure Therapy (ENPT)

An open-pore polyurethane sponge drainage (OPD), the commercially available Eso-SPONGE^®^ System (B. Braun Melsungen AG, Melsungen, Germany), was used for endoluminal placement under endoscopic view. Endoscopic placement was performed via the OPD with a suture loop, as illustrated in Figure 2a(*). Re-endoscopies for monitoring and changing of the device were performed every three to five days. 

The open-pore film drainage (OFD) for endoluminal therapy was handmade, as previously described by G. Loske et al. [16], by wrapping a thin open-pore double-layered drainage film (Suprasorb^®^ CNP, Drainage Film; Lohmann & Rauscher International GmbH & Co.KG, Rengsdorf, Germany) around the distal end of a gastric tube or the gastric segment of a nasojejunal feeding tube (Freka^®^ Trelumina, Fresenius Kabi Deutschland GmbH, Bad Homburg, Germany). Sutures (Mersilene^TM^_,_ Polyester, 4 Ph. Eur., Ethicon^®^, Norderstedt, Germany) were used for fixation of the drainage film around the tube. Drain insertion took place via nasal positioning and endoscopic guiding with a grasper. The handmade OFD on an intestinal feeding tube is shown in Figure 2b. Venting tubes in trilumen enteral feeding tubes had to be closed for ENPT (shown with a yellow clamp #). Re-endoscopies were performed every five to seven days. Figure 3 shows the endoscopic findings after implementation of an OFD.

For both ENPT devices, a continuous vacuum of −125 mmHg was generated with electronic vacuum devices (KCI V.A.C. Ulta or V.A.C. Freedom; KCI USA Inc., San Antonio, TX, USA). The use of an electronic vacuum pump ensured continuous suction and, in the event of an error message (insufficient suction or occlusion), re-endoscopy was able to be performed outside the interval. Therapy was terminated after endoscopic evaluation of the anastomosis. ENPT was finalized when a complete resolution of the perforation occurred. Stent removal was selectively conducted after 4–6 weeks, and anastomosis was re-examined.

### 2.5. Database

Analysis was performed using SPSS v. 24.0.0.1 (IBM, Armonk, NY, USA). Data were presented as means ± SD and median (range). The *t*-test was used to compare continuous variables, and the Fisher’s exact test for analysis of categorical data. A p value less than 0.05 was considered to show differences of statistical significance.

## 3. Results

In sum, 14 patients with EJAL following oncological gastrectomy were identified in our database (five females; mean age of 62 ± 13.97 years) and were treated with ENPT (*n* = 9) or stent placement (*n* = 5). Patients’ characteristics are presented in Table 1. The mean time of the EJAL diagnosis was day 6.00 ± 2.49 (ENPT) and day 9.2 ± 4.32 (stent; *p* = 0.3) after surgery. The symptoms that led to a diagnosis were respiratory abnormalities, conspicuous secretion via drains, fever, and elevated CRP or leukocytes in the blood sample. If an anastomotic leakage was suspected, a CT scan and an endoscopy were performed for confirmation. A leakage was detected in the CT scan in 13 of 14 patients, and the mean elevated CRP at the time of diagnosis of EJAL was 21.98 mg/dL ± 9.23. After the diagnosis of EJAL and the start of the endoluminal therapy, most patients were treated and observed in the ICU. There was no difference in ICU stay (ENPT: 4.78 ± 6.8 days and stent: 5 ± 7.6 days, *p* = 0.88), as well as no difference in hospital stay for both groups (ENPT: 38.11 ± 16.46 and stent: 30 ± 5.4, *p* = 0.36).

The index endoscopy findings varied significantly and included circumscribed insufficiencies, large leakages with the secretion of putrid fluids, and fibrin-coated anastomosis with exposed clamps. There was no difference between the two groups.

The first treatment for all patients with EJAL was either endoluminal ENPT (*n* = 9) or primary endoscopic fully covered stent implantation (*n* = 5). To address distant extraluminal collection and sepsis control, surgical debridement and drainage were performed in 13 of 14 patients. No anastomotic re-do or leakage-suture was performed in our cohort. Accordingly, therapy success was 88% in ENPT and 100% in the stent group. The patient with failed ENPT as a first-line endoscopic therapy was further treated with a fully covered metal stent.

The number of required endoscopies was significantly lower in patients with primary stenting (6 ± 3.5 in ENPT, 2.4 ± 0.55 in stent; *p* = 0.048). Oral feeding was immediately possible after stenting, while patients who received ENPT needed an enteral feeding tube or parenteral nutrition. The treatment characteristics are summarized in Table 2.

In one patient, a stent dislocation occurred three weeks after implantation. The stent was removed endoscopically, the EJAL was found to be healed up, and the therapy was terminated successfully.

Subsequently, endoscopic interventions for the treatment of an anastomotic stenosis were not necessary in the analyzed patients. Adjuvant or additive chemotherapy could be performed in all patients after anastomotic healing.

## 4. Discussion

We present a comparative case series on 14 patients treated for EJAL after oncological gastrectomy with primarily endoscopic treatment strategies. Five patients were treated with stents, and nine patients were treated with ENPT. In all patients, the anastomosis was preserved, and sepsis control was achieved by primarily endoscopic and secondarily surgical treatment. There is currently no standard algorithm for the management of EJAL. Until 2018, in our clinic, patients with EJAL were treated with covered stents and then with ENPT according to our internal clinical guidelines, which are regularly updated in accordance with the literature and discussion by an interdisciplinary board. Clipping, a good option for fresh leaks with good perfusion, was not performed because of the late diagnosis of EJAL [19].

The most important prognostic factor for the successful treatment of EJAL is an early and reliable diagnosis. We used contrast-enhanced CT scans and endoscopic verification. In all cases of EJAL, the CT scan led to a diagnosis because of free-gas figures or fluid collection adjacent to the anastomotic region. Although CT scanning alone cannot rule out ischemia, an endoscopy was performed in all patients with the option to treat EJAL in the same session. The evaluation showed that the diagnosis of insufficiency can be made with a CT scan, and it is also possible to detect perianastomotic fluid collections, allowing a decision on whether an operative revision is necessary. A generous indication for endoscopy is additionally given in any case of worsening of the clinical course or suspicion of an anastomosis problem.

ENPT is an effective endoscopic and minimally invasive option for the management of anastomotic insufficiencies in the upper and lower gastrointestinal tract [11,12,13,16,20,21,22,23]. The first case reports of ENPT for the treatment of EJAL were presented by G. Loske [9] and J. Wedemeyer [13]. From these first reports until now, around 55 cases of EJAL treated with ENPT have been published. Five articles are also available with case series of 15 [24], 14 [25], 9 [26], 9 [27], and 5 [28] EJAL patients. Articles reporting on ENPT for EJAL are listed in Table 3. Most of the described ENPT case series of the upper gastrointestinal tract were in patients with mixed indications and with different regions of anastomotic leakage [14,15,19,25,26,27,28,29,30]. The advantages of ENPT are internal endoluminal drainage avoiding aggressive extraluminal digestive fluid leakage in the free abdomen, the stimulation of granulation followed by size reduction of the wound, and finally, preservation of the anastomosis. The main advantage, however, is the avoidance of risky re-operations. A possible reason for the effective intrathoracic therapy of ENPT in preventing dangerous mediastinitis in several studies may be the confined space, in contrast to the wide intraabdominal compartment.

Numerous articles favor the primary use of self-expandable fully or partially covered metal stents to close large defects, which are superior to plastic stents because of the lower risk of migration [3,32,33]. However, stents cover the defect without addressing extraluminal fluid collection. Therefore, existing fluid collections need drainage. Another known issue in stent therapy is stent dislocation and stent occlusion.

In the current analysis, in one patient, a stent dislocation occurred three weeks after implantation. The stent was removed endoscopically, the EJAL was healed, and the therapy was terminated successfully. The reason for changing the endoscopic therapy of EJAL in 2018 from stent implantation to ENPT at our institution was to avoid risky percutaneous drainage and possible re-operations as well as better sepsis control and stimulation of wound granulation.

In our analyzed patients, only one patient in the ENPT group did not need a re-do surgery. Surgery after EJAL was necessary because of intraabdominal fluid collection diagnosed by CT scan, distant to the leakage, with a high risk of not being reached by the intraluminal ENPT. Regardless of this, however, all anastomoses were preserved. Because of malnutrition and the multimodal treatment of gastric cancer, the majority of patients with EJAL after gastrectomy are critically ill, and those in our study were treated in the ICU/IMC. The analyzed patients had a prolonged postoperative course with invasive ventilation or closer monitoring. Intensive care support after the diagnosis of EJAL was required in all patients, with no difference in the length of stay between the two groups. Patients who received stent therapy had a trend toward a shorter hospital stay and had a lower number of endoscopic interventions compared to those who received ENPT therapy. Our expectation that ENPT would be superior to stent therapy for both wound healing and hospital stay was not met. On the contrary, ENPT required more endoscopic intervention with a longer gap of enteral feeding, especially for oral food intake. In addition to the primary outcome parameters of mortality, sepsis control, oral feeding, and hospitalization time, the further oncological course is also important for patients. The majority of patients treated with neoadjuvant chemotherapy need adjuvant completion chemotherapy [5]. Therefore, it is important for EJAL therapy to also focus also on the multimodal oncological strategy, with the opportunity for timely treatment. After recovery, all analyzed patients continued chemotherapy in an adequate time frame, and there was no difference between the two groups.

We are aware of the limitations of this retrospective, monocentric, and non-randomized case series. The single-center design also represents a source of bias. However, to the best of our knowledge, this is the first comparison of two endoscopic treatment options as first choices in the management of EJAL patients exclusively after oncological gastrectomy. Fourteen patients with EJAL is a not a representative number of patients, so consecutive multicentric studies for this topic are needed. The data do not yet allow us to determine whether patients with EJAL would benefit more from one of the two endoscopic therapy options.

Our study showed good results for the primary endoscopic treatment of EJAL. A re-do of surgery with interventions at the anastomosis was avoided in all patients. Overall, there was no difference for ENPT compared to stent therapy. The advantages of the use of a stent are the possibility of early oral food intake, fewer endoscopic interventions, and earlier discharge. As a result of our findings, stent therapy in patients with EJAL after gastrectomy is in vogue again and has limited vacuum therapy to special cases, with the option to combine both procedures.

## 5. Conclusions

Primarily endoscopic therapeutic strategies such as stent placement and ENPT are promising options for patients with EJAL after oncological gastrectomy. Both the presented therapeutic strategies have their advantages and can be used safely. The regular evaluation of healing should be performed in order to adjust the strategy or to indicate surgery if necessary.

## Figures and Tables

**Figure 1 cancers-14-02982-f001:**
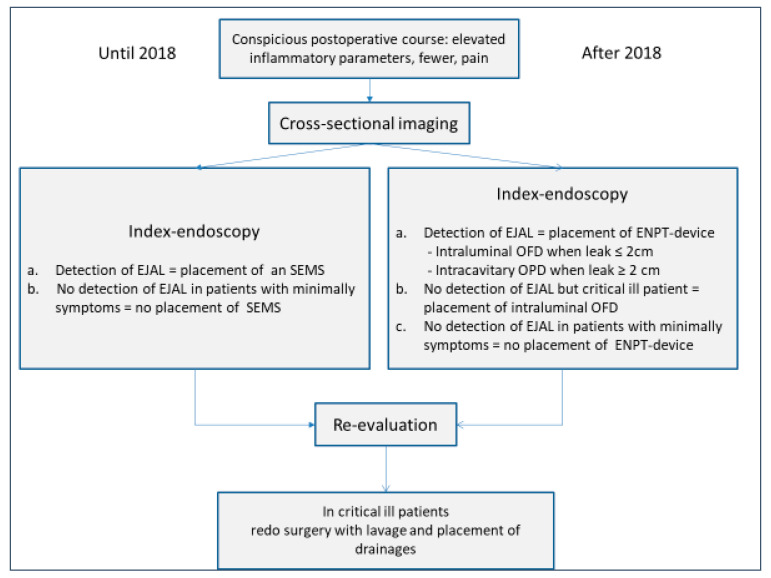
Flowchart of diagnosis and endoscopic treatment until and after 2018. EJAL = esophagojejunal anastomotic leak, SEMS = self-expandable metal stent, ENPT = endoscopic negative pressure therapy, OFD = open-pore film drainage, OPD = open-pore polyurethane sponge drainage.

**Figure 2 cancers-14-02982-f002:**
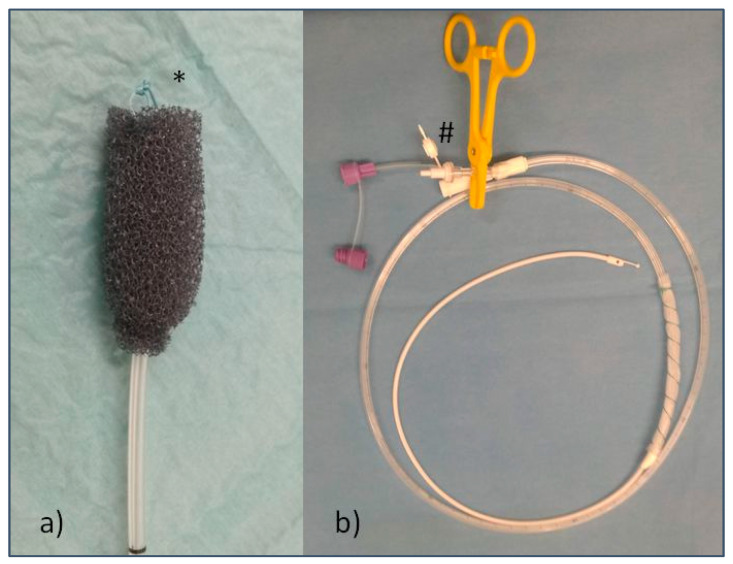
Illustrations of the two ENPT devices used; (**a**) OPD (open-pore polyurethane sponge) with suture loop (*) for forceps maneuver; (**b**) OFD (open-pore film drainage), prepared Freka^®^ Trelumina tube; the gastric part is wrapped with CNP Drainage Film and fixed with suture, and the venting tube is clamped (#).

**Figure 3 cancers-14-02982-f003:**
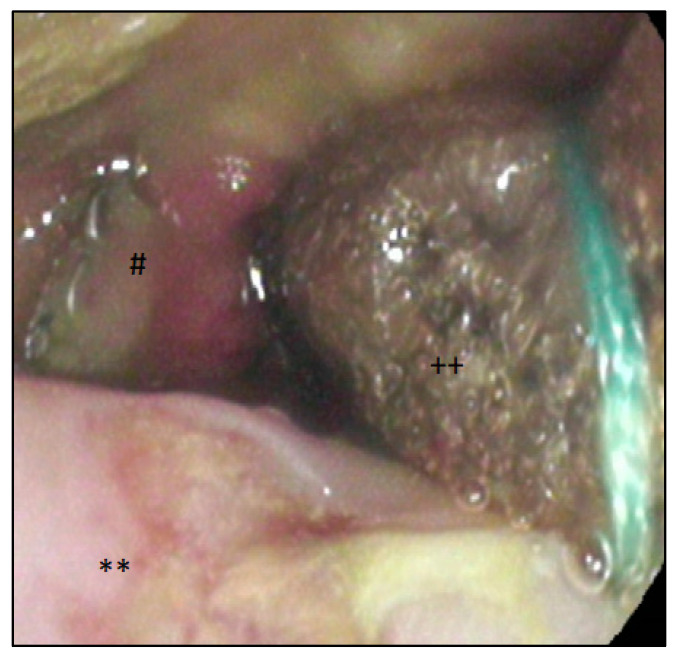
Implanted OFD endoluminal via the anastomosis; **: esophagus; #: anastomosis with exposed clamps; ++: OFD with CNP film wrapped on an enteral feeding tube in position of the EJAL.

**Table 1 cancers-14-02982-t001:** Patients’ characteristics.

Characteristics	ENPT	Stent	*p*
Number of treated patients	9	5	-
Sex (female)	4	1	-
Median age (years) [Min–Max]	60 (36–79)	61 (21–79)	0.7606
Neoadjuvant treatment (*n*) (%)	7 (77.78%)	3 (80%)	0.5804
BMI (kg/m^2^)	27 ± 9.4	29.5 ± 5.1	0.6115
Previously presented risk factors			
Obesity (BMI > 35 kg/m^2^) (*n*)	1	1	-
High age (>70 years) (*n*)	3	1	-
Previous chemotherapy (*n*)	6	3	-
Diabetes (*n*)	2	2	-
Cachexia (BMI < 15 kg/m^2^) (*n*)	0	0	-
Nicotine abuse (*n*)	5	1	-
Oncological resection (*n*)			
Complete gastrectomy with D2 lymphadenectomy	3	1	-
Combined transhiatal distal esophagectomy and gastrectomy	6	4	-
with D2 lymphadenectomy			
Histopathological resection state (*n*)			0.2582
R0	6	5	
R1	3	-	
Diagnosis of EJAL on day after surgery (Mean ± SD)	6 ± 2.55	9.2 ± 4.32	0.2981
WBC/µL at time of diagnosis of the EJAL (Mean ± SD)	11,892 ± 5507	10,058 ± 5862	0.5698
Level of CRP in mg/dl at time of diagnosis of the EJAL (Mean ± SD)	21.81 ± 11.48	22.28 ± 4.04	0.9319

ENPT = endoscopic negative pressure therapy; BMI = body mass index; D2 lymphadectomy = paragastric and suprapancreatic lymphadenectomy; WBC = white blood cells; CRP = C-reactive protein; EJAL = esophagojejunal anastomotic leak.

**Table 2 cancers-14-02982-t002:** Therapeutic data.

Characteristics	ENPT *n* = 9	Stent *n* = 5	*p*
Mean time interval between oncological gastrectomy and endoscopic diagnosis and treatment start (days)	6.00 ± 2.49	9 ± 4.18	0.1228
Number of patients requiring invasive ventilation (*n*)	9 (100%)	3 (60%)	0.1099
Mean duration of required ventilation (days)	5.56 ± 4.09	1 ± 0.83	0.0343
Mean therapy duration (days)	14.78 ± 9.66	26 ± 7.6	0.0626
Enteral feeding via (*n*)			
- Enteral Tube	9	0	<0.01
- Oral	0	5	-
Number of endoscopies needed per patient (*n*)	6.0 ± 3.52	2.4 ± 0.55	0.0462
Number of patients requiring combined surgery (*n*)	8 (88.89%)	5 (100%)	1.0
ICU stay needed in patients (*n*) (%)	9 (100%)	4 (80%)	0.3571
Mean duration of ICU stay (days)	4.78 ± 6.8	5 ± 7.6	0.8816
Mean duration of hospital stay (days)	38.11 ± 16.46	30 ± 5.4	0.3622
Treatment success (*n*) (%)	8 (88.89%)	5 (100%)	1.0

ENPT = endoscopic negative pressure therapy; ICU = intensive care unit.

**Table 3 cancers-14-02982-t003:** Literature review focused on ENPT for EJAL.

Author	Year of Publication	Period Analyzed	Patients Treated for EJAL (*n*)	Patients Treated for Leaks of the UGI (*n*)	ENPT Success in EJAL Patients (*n*)
Bludau et al. [24]	2018	October 2010–January 2017	15	77	Not specified
Brangewitz et al. [25]	2010	January 2010–July 2011	14	32	Not specified
Kuehn et al. [28]	2012	March 2011–May 2012	5	9	Not specified
Kuehn et al. [14]	2016	March 2011–March 2015	Unspecified	21	Not specified
Laukoetter et al. [26]	2017	December 2011–December 2015	9	52	Not specified
Loske et al. [9]	2009	2009	1	1	1
Mencio et al. [31]	2017	July 2013–December 2016	Unspecified	36	Not specified
Schorsch et al. [27]	2014	November 2006–October 2013	9	35	Not specified
Wallstabe et al. [12]	2010	2010	1	1	1
Wedemeyer et al. [13]	2008	2007	1	2	1
**Total:**			**55**	**266**	

ENPT = endoscopic negative pressure therapy; EJAL = esophagojejunal anastomotic leak; UGI = upper gastrointestinal tract.

## Data Availability

Not applicable.

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
