# Peer review of "Comparison of Two Endoscopic Therapeutic Interventions as Primary Treatment for Anastomotic Leakages after Total Gastrectomy"

_cancers, 2022, doi:10.3390/cancers14122982_

Round 1

Reviewer 1 Report

The authors compared the two treatment strategies in a treatment for anastomotic leakages after total gastrectomy. They concluded that good clinical results preserving the and providing sepsis control were achieved in all patients. Stent therapy resulted in anastomosis healing with lower number of endoscopies, shorter time of hospitalization, and fast oral nutrition.

Although study is interesting there are several major objections in study design and methodology. My concerns are as follows:

  1. Abstract – The authors stated ‘’ In the stent group 2.4 endoscopies were performed…’’ If that was a mean or median they should state mean of 2.4 endoscopies were performed… The same goes for ‘’endoscopic negative pressure therapy 6.00 endoscopies…
  2. Methodology is poor and not informative. Many important issues have not been presented. First, study design is poorly presented. Each variable measured in any patient should be described.
  3. Methodology – Please indicate primary and secondary outcomes of the study in a separate paragraph.
  4. Methodology – The authors did not present clear inclusion / exclusion criteria of the study
  5. The authors stated that the study was approved by institutional ethics committees. Please add the date of approval, next to the reference number.
  6. Statistical analysis – Which statistical test was used to test normality of distribution of the data? Also it is unclear which p-value was considered as statistically significant.
  7. Sample size – A sample of 14 patients is too weak to draw any serious conclusions. In line to that the comparisons between two groups are mostly inadequate. The authors included the patients between January 2014 and December 2019. I wonder why the patients from the last 2 and a half years were not included.
  8. Figure 1 – Please add a description for each abbreviation used in Figure in the legend, e. g. EJAL, SEMS, ENPT …
  9. It is unclear how the decision on type of endoscopic technique was made, preference of operating surgeon, or the patients before 2018 received one intervention and after 2018 another? Please explain.
  10. The authors stated: ''In accordance with ethical approval, informed consent was obtained from all participants''. How it was possible as this was a retrospective study?
  11. Tables – Each abbreviation mentioned in a Table should be explained in the legend of each Table, even if it was previously mentioned in text.
  12. Discussion is poor and should be improved. Discussion section needs to be re-written/re-arranged. There is much repetition from existing literature. The authors should focus on results from the main objectives of the study. Write in four sequential paragraphs (without headings); (i) summary (not data) of findings from present study; (ii) logical and coherent comparison with existing literature with focus of comparison on main objective(s); (iii) limitations of the study; and (iv) Implications for practice/policy/research with a concluding statement.
  13. The authors did not even mention all limitations of their study in discussion. Single center design should be mentioned as well as all possible sources of bias.
  14. Quality of English should be improved. Manuscript should be edited for the language from a native English speaker or professional language editing service to improve the grammar and readability.

Author Response

The authors compared the two treatment strategies in a treatment for anastomotic leakages after total gastrectomy. They concluded that good clinical results preserving the and providing sepsis control were achieved in all patients. Stent therapy resulted in anastomosis healing with lower number of endoscopies, shorter time of hospitalization, and fast oral nutrition.

Although study is interesting there are several major objections in study design and methodology. My concerns are as follows:

R1: Abstract – The authors stated ‘’ In the stent group 2.4 endoscopies were performed…’’ If that was a mean or median they should state mean of 2.4 endoscopies were performed… The same goes for ‘’endoscopic negative pressure therapy 6.00 endoscopies…

#R1.1: Thanks for this advice, we changed the sentences: In the stent group mean number of endoscopies was 2.4 endoscopies, mean duration of therapy was 26 days, mean time of hospitalization was 30 days. In patients treated with endoscopic negative pressure therapy mean number of endoscopies was 6.0, mean days of therapy duration was 14.78 and mean days of hospitalization was 38.11, respectively.

Methodology is poor and not informative. Many important issues have not been presented. First, study design is poorly presented. Each variable measured in any patient should be described.

R1.2 Methodology – Please indicate primary and secondary outcomes of the study in a separate paragraph.

#R1.2: Thanks for this note. As primary outcome parameter, we investigated the success rate of endoscopic interventions on the healing of the anastomosis. Secondary outcomes were need of invasive ventilation, time at ICU, therapy duration, possibility of oral and enteral feeding, number of intervention (surgery and endoscopy) and hospital stay. We inserted this information into the methodology section.

R.1.3 Methodology – The authors did not present clear inclusion / exclusion criteria of the study

#R1.3: This information is also included into the reviewed submission. Inclusion criteria were adult patients after elective gastrectomy in a curative approach with an EJAL and primary endoscopic intervention between 2014 and December 2019. Excluded were patients in a palliative setting and with primary surgical re-do anastomosis after diagnosis of EJAL.

The authors stated that the study was approved by institutional ethics committees. Please add the date of approval, next to the reference number.

#R.1.4: We inserted the missing information.

Statistical analysis – Which statistical test was used to test normality of distribution of the data? Also, it is unclear which p-value was considered as statistically significant.

#R1.5: Thanks for this note. Following sentence is included now: Analysis was performed using SPSS v. 24.0.0.1 (IBM, Armonk, NY, USA). Data were presented as means ± SD and median (range). T-Test was used to compare continuous variables, and the Fisher’s exact test for analysis of categorical data. A p value less than 0.05 was considered to show differences of statistical significance.

Sample size – A sample of 14 patients is too weak to draw any serious conclusions. In line to that the comparisons between two groups are mostly inadequate. The authors included the patients between January 2014 and December 2019. I wonder why the patients from the last 2 and a half years were not included.

#R1.6: Thanks for this advice. Because of COVID-19-pandemic many changes took place. We had a lower number of gastrectomies for cancers in the last two years. The follow-up examinations were done only in a low number of patients because of COVID-19 regulatories. These changes are not content of the present manuscript, therefore patients were included until 2019.

Figure 1 – Please add a description for each abbreviation used in Figure in the legend, e. g. EJAL, SEMS, ENPT …

#R1.7: We included the missing abbreviations.

It is unclear how the decision on type of endoscopic technique was made, preference of operating surgeon, or the patients before 2018 received one intervention and after 2018 another? Please explain.

#R1.8: Please see the decision tree in Figure 1. All patients with primarily endoscopic therapy for EJAL before 2018 were treated with stents. After 2018 they were treated with ENPT. Decision for endoscopic primarily therapy was done by the operating surgeon and the interventional endoscopist. 

The authors stated: ''In accordance with ethical approval, informed consent was obtained from all participants''. How it was possible as this was a retrospective study?

#R1.9: At our center, patients are informed before the start of therapy about the possible use of the collected data in the context of later evaluations. In this case, only data generated during the regular hospital stay will be used. A respective consent was given for all included patients.

R 1.10: Tables – Each abbreviation mentioned in a Table should be explained in the legend of each Table, even if it was previously mentioned in text.

#R1.10: Thanks for this note, we included the missing information.

R 1.11: Discussion is poor and should be improved. Discussion section needs to be re-written/re-arranged. There is much repetition from existing literature. The authors should focus on results from the main objectives of the study. Write in four sequential paragraphs (without headings); (i) summary (not data) of findings from present study; (ii) logical and coherent comparison with existing literature with focus of comparison on main objective(s); (iii) limitations of the study; and (iv) Implications for practice/policy/research with a concluding statement.

#R1.11:  Thanks for this advice. We revised the discussion section completely.

R1.12: The authors did not even mention all limitations of their study in discussion. Single center design should be mentioned as well as all possible sources of bias.

#R1.12: Thanks for this note. We revised the limitation section.

R 1.13: Quality of English should be improved. Manuscript should be edited for the language from a native English speaker or professional language editing service to improve the grammar and readability.

#R1.13: We revised the manuscript with an editing service.

Reviewer 2 Report

I read with interest the manuscript "comparison of two endoscopic therapeutic interventions as primary treatment for anastomotic leakage after total gastrectomy". I believe that this study does add something despite the fact that there are quite numerous reports on the use of stents in UGI leakages. I have few comments

  1. Please explain based on which parameters you have decided which patient would be treated with a stent or ENPT
  2. please report what was the method of the esopago-jeujonostomy anastomosis in the index surgery.
  3. if there were two methods (circular vs linear) or more types of anastomosis in the index surgery did it play a role in the decision between stent or ENPT.
  4. Please explain why only 4 patients had a D2 gastrectomy
  5. Did you have any other patients in this time frame that had referred straight to surgery following a leak without trying one of the endoscopic methods
  6. did you have any stent migration episodes
  7. did you look/find any difference between patients in their ability to tolerate the ongoing treatment in each of the two methods? from my experience some patients had a problem to tolerate the stent and we had to take it earlier than we expected
  8. How many of the patients developed dysphagia after treatment and was it more common in any of the groups.
  9. what were the criteria to end the treatment? in other words based on which criteria you have decided to take out the stent or to stop the ENPT
  10. Please report on the cancer recurrence rate in this group as we know that postoperative leaks are a risk factor for disease recurrence.

Author Response

I read with interest the manuscript "comparison of two endoscopic therapeutic interventions as primary treatment for anastomotic leakage after total gastrectomy". I believe that this study does add something despite the fact that there are quite numerous reports on the use of stents in UGI leakages. I have few comments

R2.1: Please explain based on which parameters you have decided which patient would be treated with a stent or ENPT

#R2.1: As mentioned in the introduction part: Since 2018 ENPT is the primary endoscopic therapeutic option for EJAL at our institution, after years of stenting using different self-expandable metal stents (SEMS). The reason for changing our approach to primary ENTP for EJAL was the expectation to establish a better wound healing with simultaneous sepsis control and lower re-operation rate. Please find the decision tree in Figure 1.

R2.2: Please report what was the method of the esopago-jeujonostomy anastomosis in the index surgery. If there were two methods (circular vs linear) or more types of anastomosis in the index surgery did it play a role in the decision between stent or ENPT.

#R2.2: In both groups, eosophago-jejunostomy was performed end to side with circular stapler without a jejunal pouch. The blind end of the jejunum was closed with linear stapler.

R2.3: Please explain why only 4 patients had a D2 gastrectomy

#R2.3: Thanks for this advice, of course were patients with transhiatal extended gastrectomy also operated with D2 gastrectomy. We included this information into the Table 2. 

R2.4: Did you have any other patients in this time frame that had referred straight to surgery following a leak without trying one of the endoscopic methods

#R2.4: Patients with EJAL and primary surgical re-do of the anastomosis were excluded. We inserted this information.

R.2.5: Did you have any stent migration episodes?

#R2.5: In the current analysis in one patient a stent dislocation occurred three weeks after implantation. The stent was removed endoscopically, the EJAL was healed up and the therapy could be terminated successfully.

R2.6: Did you look/find any difference between patients in their ability to tolerate the ongoing treatment in each of the two methods? from my experience some patients had a problem to tolerate the stent and we had to take it earlier than we expected

#R2.6: Thanks for this question. We have no data for this topic. There was only one stent removal because of dislocation. For patient’s discomfort was no earlier stent removal necessary in the analyzed time period.

R2.7: How many of the patients developed dysphagia after treatment and was it more common in any of the groups.

#R2.7: Thanks for this interesting question, at this time we have no dysphagia questionnaire answered by patients. We will start with this data acquis for this question yet.

R2.8: What were the criteria to end the treatment? in other words based on which criteria you have decided to take out the stent or to stop the ENPT

#R2.8: Thanks for this advice. ENPT was finalized when a complete resolution of the perforation was seen. Stent removal was electively done after 4 weeks and anastomosis was re-examined.

R2.9: Please report on the cancer recurrence rate in this group as we know that postoperative leaks are a risk factor for disease recurrence.

#R2.9: Due to the retrospective design, long-term data from all patients does not exist. In 4 cases (ENPT) a recurrence of gastric cancer is noted in our database, in the stent group one recurrence is described in our database. Due to the lack of data on some patients, we did not draw any conclusions from this fact.

Reviewer 3 Report

The authors present the outcomes from a retrospective, non randomized, single center cohort of patients who were endoscopically treated for anastomotic leak after gastrectomy during a 5-year period. The authors present outcomes for a total of 14 patients and based on their outcomes they conclude that good clinical results preserving the and providing sepsis  control was achieved in all patients while stent therapy resulted in anastomosis healing with lower number of endoscopies, shorter time of hospitalization, and fast oral nutrition.

This is an interesting study focusing on minimally invasive techniques addressing anastomotic leaks following gastrectomy for ?cancer.

A number of issues that need to be addressed by the authors:

- It would be interesting to know how many patients leaked (%) compared to how many patients were operated during the study period.

- It should be made clear in the methods that all patients were operated for malignancy. Also R status should be given.

- The limitations section is rather poor. "We are aware of the limitations of this retrospective case series" is simply unacceptable, all those limitations need to be provided to the readership.

- What about longer term outcomes? Did any of the patients get readmitted? Did they experience stenosis during follow up? These techniques might have good outcomes but on the long-run may be hazardous to the patients, this needs to be clarified by the authors in order to highlight the good efficacy of their approach.
  - Some endoscopic images of sponge placement vs leak could be useful for the readership  

Author Response

The authors present the outcomes from a retrospective, non-randomized, single center cohort of patients who were endoscopically treated for anastomotic leak after gastrectomy during a 5-year period. The authors present outcomes for a total of 14 patients and based on their outcomes they conclude that good clinical results preserving the and providing sepsis control was achieved in all patients while stent therapy resulted in anastomosis healing with lower number of endoscopies, shorter time of hospitalization, and fast oral nutrition.

This is an interesting study focusing on minimally invasive techniques addressing anastomotic leaks following gastrectomy for cancer.

A number of issues that need to be addressed by the authors:

R3.1 It would be interesting to know how many patients leaked (%) compared to how many patients were operated during the study period.

#R3.1: Thanks for this question. We think that this information is not relevant for this manuscript, because we only analyze patients with primary endoscopic therapeutic interventions for EJAL. The focus of this manuscript is to determine which of the methods presented is more advantageous.

R3.2: It should be made clear in the methods that all patients were operated for malignancy. Also, R status should be given.

#R3.2: Thanks for this note. Patients in the stent group were all classified as R0. Patients in the ENPT group had following histopathological R-state: n=3 R1, n=6 R0. We inserted this information into Table 1.

R3.3: The limitations section is rather poor. "We are aware of the limitations of this retrospective case series" is simply unacceptable, all those limitations need to be provided to the readership.

#R3.3: Thanks for this note. We revised the limitation section.

R3.4: What about longer-term outcomes? Did any of the patients get readmitted? Did they experience stenosis during follow up? These techniques might have good outcomes but on the long-run may be hazardous to the patients, this needs to be clarified by the authors in order to highlight the good efficacy of their approach.

#R3.4: No patient was re-admitted because of stenosis or renewed EJAL. Four patients in ENPT group and one in the Stent group had and gastric cancer recurrence. Due to the lack of data on some patients, we did not draw any conclusions from this fact.

R3.5: Some endoscopic images of sponge placement vs leak could be useful for the readership  

#R3.5: Thanks for this advice. We included Figure 3.

Round 2

Reviewer 1 Report

The authors revised the manuscript according to the reviewer’s suggestions. After revisions the manuscript has been improved (mostly in regards to minor objections) but the main objection still remains. The sample size is too weak to draw any serious conclusion. The authors performed analysis between the two groups. First group had only 5 patients and the second only 9. Majority of analyses are inappropriate and there is a significant source of bias. The only way to improve this manuscript is to enlarge sample size and resubmit the manuscript after 2-3 years when the sample size would be more representative.

Also, the authors did not respond to which statistical test was used to test normality of distribution. As I can see that was not performed, so this is also one of possible sources of bias.

I am sorry but I cannot support publication of the manuscript with poor design and sample size.

Reviewer 2 Report

the authors responded adequately to my comments

Reviewer 3 Report

The authors have adequately addressed my remarks